# Assessing the global implications of the COVID-19 pandemic on the cervical cancer elimination initiative

Anisha M. Loeb[1], Patti Gravitt[2], Allison Frank[2], Douglas M. Puricelli Perin[3], Kalina Duncan[2], Linda Eckert[4,5], Maribel Almonte[6,7], Nathalie Broutet[8], Joseph Rodman[9], Oyeleke A. Oyebamiji[9], Prajakta Adsul[9,10]*

**1** Department of Epidemiology, School of Public Health, University of Washington, Seattle, Washington, United States of America, **2** Center for Global Health, National Cancer Institute, Rockville, Maryland, United States of America, **3** Clinical Monitoring Research Program Directorate, Frederick National Laboratory for Cancer Research, Frederick, Maryland, United States of America, **4** Department of Obstetrics & Gynaecology, University of Washington, Seattle, Washington, United States of America, **5** Department of Global Health, School of Public Health, University of Washington, Seattle, Washington, United States of America, **6** Department of Non-Communicable Diseases, World Health Organization, Geneva, Switzerland, **7** Early Detection, Prevention and Infections Branch, International Agency for Research on Cancer, Lyon, France, **8** Department of Sexual and Reproductive Health and Research, World Health Organization, Geneva, Switzerland, **9** Cancer Control and Population Sciences, University of New Mexico Comprehensive Cancer Center, Albuquerque, New Mexico, United States of America, **10** Department of Internal Medicine, University of New Mexico, Albuquerque, New Mexico, United States of America

* padsul@salud.unm.edu

## Abstract

The COVID-19 pandemic disrupted many public health programs; understanding these disruptions is critical for directing future resources. In a project studying the implementation of human papillomavirus (HPV) testing-based cervical cancer screening, we queried about the impact of the pandemic on screening programs globally. In consultation with World Health Organization's Regional Advisors, program managers, government officials, and clinicians involved in the implementation of HPV testing-based cervical cancer screening programs were invited to participate in semi-structured, in-depth, interviews. Interview notes and transcripts were used for inductive analysis, focusing on responses to the impact of COVID-19 pandemic on screening programs. Thirty-two interviews were conducted with participants between the age of 29 and 61 years, representing programs from 25 countries. Six key themes were noted. Regarding disruptions, (1) the entire cancer continuum was affected, leading to delays or, in some cases even cessation of vaccination, screening, and treatment programs; and (2) a heightened sense of fear around contracting and transmitting COVID-19 shifted government priorities and impacted healthcare delivery. Nonetheless, participants noted constructive ways in which programs leveraged the impact of the pandemic: (1) at the community level, participants were able to leverage an increased understanding and acceptance surrounding the importance

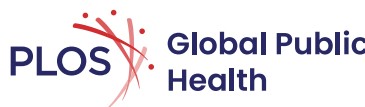

**Data availability statement:** All relevant data are within the paper.

**Funding:** This project has been funded in whole or in part with federal funds from the National Cancer Institute, National Institutes of Health, under Contract No. 75N91019D00024. The content of this publication does not necessarily reflect the views or policies of the Department of Health and Human Services, nor does mention of trade names, commercial products, or organizations imply endorsement by the U.S. Government. In addition, this project has been funded by the UNDP (United Nations Development Program)–UNFPA (United Nations Population Fund)–UNICEF–WHO–World Bank Special Program of Research in Human Reproduction (HRP) (contract to PA; WHO PO # 202833289). The funders had no role in the design of the study and analysis of data.

**Competing interests:** The authors have declared that no competing interests exist.

of preventive health behaviors; (2) for HPV-testing: molecular laboratories became well-equipped with better technician training, increasing overall HPV-testing capacities; (3) the pandemic promoted virtual healthcare systems; and (4) for planning: shutdowns allowed time to plan for program scale-up. In many ways, the pandemic response provided evidence supporting countries' abilities to mobilize resources. While disruptions were noted, the pandemic provided implementers with opportunities to strengthen screening programs, which should be further assessed in future sustainability research on cervical cancer prevention and control.

## Introduction

Cervical cancer (CC) continues to be commonly diagnosed and fatal among women in low- and middle-income countries (LMICs) [1]. However, it is preventable through interventions such as human papillomavirus (HPV) vaccination and regular CC screening followed by treatment of positives. Women are vulnerable to HPV infection in the cervix across their lifespan [2], and it is essential that they are screened regularly accompanied by the recommended follow-up and treatment for those who are found to have screening abnormalities. In 2021, the World Health Organization (WHO) put out a recommendation for the CC screening, which proposed several screening and treatment algorithms, with a strong focus on using HPV DNA testing as the primary screening test due to its high accuracy [3]. In response, many countries have adopted and, in some cases, implemented HPV DNA screening programs following the WHO recommendations; however, there is limited guidance for implementation of population-based HPV DNA screening and treatment programs.

Moreover, the COVID-19 lockdown in March 2020 caused significant disruptions to health systems worldwide. Due to the surge of COVID-19 cases, many health systems were unable to focus on cancer screening and treatment [4]. This led to programmatic shutdowns, decreasing CC screening rates by 52% globally with reductions in rates of colposcopy and invasive CC diagnoses [5–11]. Public interest in cancer and other health conditions lessened as media coverage highlighted COVID-19 as the sole health threat, and Google search trends showed decreases in searches related to cancer screenings [12].

The COVID-19 pandemic intensified barriers to seeking sexual and reproductive health care. There were severe impacts to financial and human resource availability, including supply chain disruptions, reduced programmatic funding, a shortage of healthcare workers, increased healthcare costs, lower appointment availability, and the inability of patients to afford necessary medical services [5,13–15]. The lack of resources encouraged physicians to adopt a 'risk-based approach' in providing care, prioritizing the most vulnerable patients [14]. However, defining vulnerability became difficult when physicians had to consider the severity of the disease versus the number of life-years that the treatment might save. Additionally, community members lost trust in the healthcare system [14,16]. This distrust coupled with fear of contracting COVID-19 and pre-existing transportation barriers further limited and discouraged patients from seeking healthcare services [13–15,17,18].



At the same time, the pandemic also highlighted potential strategies to enhance healthcare delivery, particularly in relation to CC screening. The reduction of in-person clinic visits has led to a greater acceptance of telemedicine visits among providers and patients, which helps to reduce transportation barriers and scheduling constraints for many patients [14,15,18]. HPV testing allows patients to conduct self-sampling at home, which has expanded service delivery [8,14,18]. Companies that conduct microbiology testing reported increases in production capacity and a rise in the set-up of testing platforms to address COVID-19, which is expected to be useful for expanding HPV testing post-pandemic [19]. To track COVID-19 cases, many countries opted for health registries, that can be linked to national health information systems and be leveraged for other diseases such as CC [8]. Lastly, the pandemic has facilitated strong collaboration and coordination between health and governmental sectors which can aid in policy and financial support for preventive health programs [4,5,17,20].

The study is a part of a larger initiative that aimed to generate practice-based evidence for strategies and approaches that support the implementation of HPV DNA-based screening and treatment programs that are ongoing across the globe guided by the Exploration, Preparation, Implementation, and Sustainment Framework [21]. Since these interviews took place amidst the COVID-19 pandemic, there were significant discussions around the impact of the pandemic on the CC screening and treatment programs. In this paper, we characterize the impacts of COVID-19 on pre-existing CC screening programs in LMICs using data from the interviews. The purpose of this study is to assess strategies that were used to address the impact of COVID-19 on CC screening programs in LMICs and how they could be leveraged to improve the functioning and preparedness of these programs for continuity and resilience amidst future public health emergencies.

## Methods

### Recruitment

All study activities were initiated after securing appropriate ethical approvals from the World Health Organization Human Reproduction Program (HRP) and the University of New Mexico's Human Research Protections Office [Study ID: 21–398]. After consulting with WHO regional advisors and the study team's professional network, we connected with 84 practitioners involved in CC screening programs from various countries across the globe. The study team, along with WHO regional advisors, sent email communications to all 84 health practitioners to describe the initiative, request and confirm their interest in participation, and schedule an interview. No participant compensation was offered or provided for this study. Written consent was obtained when the interview was scheduled. If, for some reason, the participant did not have time to complete the written consent, verbal consent was obtained with the interviewer as the witness on the recorded Zoom calls.

### Data collection

Two study team members with implementation science and qualitative research expertise, PA and PG, conducted the semi-structured interviews between June – September 2022. The interview questions focused on the effectiveness, sustainability, and scale-up potential of existing CC screening programs. Specific to the analyses presented in this paper, participants were asked "How has COVID affected the current implementation efforts? Were there specific changes during the pandemic that you have made in program delivery? Reflecting back on the innovative strategies that you have used for cervical cancer screening, are there changes you've made to the algorithms you used? By necessity you have had to be innovative, how many of those seem like better options to continue?" The Zoom virtual meeting platform was used to conduct the interviews, which lasted one to two hours. Simultaneous translation, which consisted of two live interpreters; one on an English channel with the interviewers and one on a French or Spanish channel with the participant, was provided for non-English speakers. There were a total of three interviews which required these efforts (two in French and one in Spanish). A supplemental notetaker attended each interview to ensure interview guide concordance and assisted the interviewer with relevant probes and clarifications as needed. All interviews were audio-recorded and transcribed verbatim with participant consent. We achieved data saturation at 32 interviews.

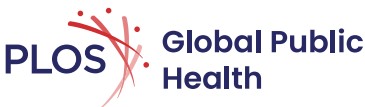

### Data analysis

The transcripts were uploaded to Dedoose qualitative software to facilitate review, organization, and analysis. The study team used a grounded theory approach to develop the COVID-19-related codes and analyze the interview transcripts and notes. Grounded theory is an inductive qualitative technique to identify themes that emerge from text [22]. Team members (AL, JR, and PA) trained in qualitative analysis read the transcripts for each interview, and independently coded them using Dedoose. During initial coding, the study team met regularly to compare codes and resolve discrepancies by the third coder. Team discussions led to the identification of six major themes from the data, and the team generated summary statements and narratives based on these themes. Collectively, these themes represent the impact of COVID-19 on the CC screening and treatment programs and were used to organize the results. Participant responses were de-identified to only include the WHO region and profession. Quotes were selected to represent each theme.

## Results

### Participants

The study team completed 32 interviews between June and October 2022. Participants were aged between 29–61 years and came from 25 countries including: 10 from the Americas region (AMR), 10 from the Africa region (AFR), four from the Europe region (EUR), one from the Eastern Mediterranean region (EMR), three from the Western Pacific region (WPR), and five from the South-East Asia region (SEAR). Out of the 25 countries, in accordance with the World Bank, two is designated as low-income, 10 as lower-middle income, 11 as higher-middle income, and two as high-income [23]. 87% of included countries have a national cervical cancer screening program and approximately half have existing HPV testing programs, introduced from 2011-2023 [24]. Participant professions included health ministry officials, program managers, clinicians, and epidemiologists.

### Theme 1: *The* COVID-19 pandemic had *an* impact *on* healthcare facilities and human resources *across the* CC continuum

Almost all participants noted that the lockdown measures implemented to combat COVID-19 caused the shutdown of healthcare facilities and public health programs, resulting in a negative impact on preventive services. The rates of HPV vaccination and CC screening rates notably decreased. Participants reported full-scale programmatic shutdowns or delays in pilot programs, setting back progress made towards CC elimination. Interestingly, there were some countries where programs continued uninterrupted by COVID-19. However, participants noted that health systems overall were unprepared for the pandemic. A primary-level health professional from the AMR region noted:

*"Because our health system …was improper. I think that, that the worse thing was when we had to stop all the programs, all the prevention programs were stopped and … we took up a bad idea to shut down all the primary care facilities. All, all were shut down at the beginning of COVID-19. So that was crazy because all, all the people were sick at the hospitals and we don't have enough hospitals, but that decision was made and all the programs …that weren't COVID-19 were shut down"*

All participants noted that to address the global emergency, financial and human resources were redirected from health programs, negatively impacting the procurement of HPV DNA tests, the availability of trained professionals, and overall testing capacity. This resulted in a backlog in diagnostic testing, which led to a subsequent backlog in treatment referrals. A lead physician in the AFR region noted:

*"Outpatient was literally closed, and they turned that area into a COVID-19 area. That meant that the screening availability was also quite reduced. Then there was a time the referrals were stopped."*

Many healthcare workers were mobilized toward COVID-19-related needs, and some were diverted to providing primary care to patients living with the virus. The diversion of these trained professionals impacted the scale and quality of care during the pandemic, but also continued to impact public health programs with the need for refresher trainings. A tertiary-level health professional from the SEAR region, not yet having implemented HPV screening, noted the impacts of COVID-19 on the workforce for VIA:

*"So, the main challenges retaining the trained staff in the same program when COVID-19 pandemic stuck. All the people trained for VIA [visual inspection using acetic acid], they are lost. They have been scattered into different health programs, so now getting them back… to the same program by itself is a challenge."*

While HPV testing was still being rolled out in some regions, participants noted that the pandemic resulted in a clear need to collect cervical samples with a limited number of healthcare workers, leading to favoring of HPV DNA tests which allowed self-sampling. Many noted that self-sampling could increase screening rates by reaching more women and many interviewees indicated patient preference towards self-sampling increasing their comfort and eliminating transportation barriers. A program director in the AFR region noted:

*"Due to COVID-19, we were, I don't want to say forced, but were kind of forced to think through a more innovative way of getting samples without having women come to the facility. This is where we added the self-sampling arm both for the women who are in the facility. They will be given the opportunity to self-sample within the facility. I agree, for me, this is the most cost-effective way of self-sampling because it eliminates first the stigma around a pelvic exam, but it also eliminates the requirement for a speculum which is a challenge in our setting, procuring as many speculums as needed."*

### Theme 2: The fear of contracting and transmitting COVID-19 shifted government priorities and impacted healthcare delivery

Several participants pointed out that the success and sustainability of programs depend on governmental or ministry support. During the pandemic, participants stressed that any focus governmental officials had on implementing CC screening programs, as well as human resources, were diverted to COVID-19 testing and care, which significantly impeded the already underfunded public health and cancer prevention programs. A public health specialist in the AFR region noted:

*"COVID-19 vaccine campaigns disrupted service delivery at health care facilities and decreased government spending for cervical cancer related services due to competing priorities."*

This disruption not only impacted preventive measures during the pandemic as reported by the interviewees but when countries were able to prioritize CC, it was difficult to resume programmatic implementation. Thus, many existing programs were affected for longer than initially thought, and screening and vaccination rates remained consistently low.

Several participants expressed widespread fear about COVID-19 among individuals seeking healthcare as well as frontline workers. A Ministry of Health representative from the AMR region noted:

*"So, I think the human resources definitely got hit hard where we had to sacrifice primary care interventions. Fear from COVID-19 was present on the demand side and the supply side. We rushed people away from the health care facilities so that they don't get infected, or they don't infect us."*

It was reported that this fear also led to physical distancing protocols, which negatively impacted the delivery of in-person healthcare services as well as community engagement and outreach. Screening rates dropped in populations

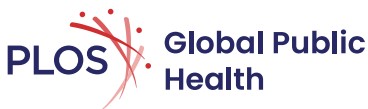

who especially rely on community campaigns for education and service delivery. A lead physician from the AFR region summarized:

> "We had this recommendation that come out for social distancing purposes, you can no longer conduct outreach, especially during the peaks. That actually resulted in reduced screening rates."

### Theme 3: Awareness around the COVID-19 pandemic led to an increase in understanding about preventive health services

Almost all participants acknowledged that the COVID-19 pandemic increased public awareness of viruses and how they are transmitted, and that such awareness led to a greater acceptance of preventive health measures such as vaccinations and early detection. A professor from the AMR region highlighted the benefits of this increased awareness and emphasized the need for further messaging to promote screening programs:

> "I think one of the critical things, I believe that …everybody by the way after COVID-19, understands what a virus is, what a virus can do, and what a molecular test is. That's a big advantage. They understand that somebody can stay with the virus. They even are asking me if they can be vaccinated because they also think about COVID-19. Unfortunately, we have to tell them that we can vaccinate only the girls for the future, but they already had the virus, so at this point what we need to do is to find that the virus is there, and we have to treat them. We're using that message, so communication is also an important thing."

As noted above, this perceived improvement in awareness among the populations may be a potential advantage for public health communications for CC prevention.

### Theme 4: Preparedness for COVID-19 testing had an impact on CC screening testing capacities

According to several participants, the laboratories used the same testing platforms for various diseases such as HPV, HIV, and TB. PCR tests for COVID-19 also leveraged these same testing platforms and thus countries were able to use preexisting testing platforms to address the pandemic. This meant that the testing capacity for CC was undercut, which led to delays in testing and increased chances of loss to follow-up. A health authority professional from the AFR region noted:

> "Oh, COVID-19 testing. Another thing is COVID-19 testing all platforms. Oh my God. That was a nightmare because all platforms were busy with COVID-19 testing. So, all the HPV testing were put on hold for like--in some places, even for two months."

However, to combat the scale of the pandemic, more individuals received training and became familiar with molecular testing platforms. Additional financial support and improvement of physical infrastructure increased the testing capacities in these laboratories. These advancements in testing capabilities can be leveraged for HPV testing, especially as the incidence of COVID-19 has slowed down. A health authority professional from the AMR region noted, candidly:

> "But, with the change of COVID-19, biomolecular testing is no longer rocket science. So there's an opportunity to think of other technologies that could be used, and rethink,… what if now we could do this dry swabs... [if] we need a PCR machine and we could do this in a regional hospital...So I think there's an opportunity to use technologies that are not going to be used, hopefully, for SARs, COVID 2 in the future, that are in a lot of these hospitals that had no biomolecular testing in the past. So, that's another opportunity."

Participants shared many lessons learned from implementing widespread COVID-19 testing that could inform future processes for HPV testing. This includes the use of barcodes to ensure quality, text messaging to communicate results linked to health information systems, and the centralization of services, as explained by an epidemiologist in the EUR region:

*"We take one positive advantage that year because that year was many connections with Directory of Public Health of the District with the Institute of Public Health. Because they collected the sample for COVID-19 and they sent all these samples in the lab of the Institute. Because the lab of the Institute of [country name removed] was the main lab for analyzing COVID-19 tests. There were many connections, so this is the positive to cover all the program in a few months, the program of the screening."*

### Theme 5: The COVID-19 pandemic revealed fragmentation of healthcare systems and also promoted a move towards virtual healthcare systems

Some participants shared concerns about the impact of the pandemic on the fragmented healthcare systems, while adding that well-functioning systems are crucial for screening programs to remain successful. An epidemiologist from the AMR region highlighted this problem, describing through a scenario of managing patients living with COVID-19 compared to other primary care services:

*"Okay, if a woman is HPV positive at the center, where will they be sent? To hospital A. Okay, these women were sent to the hospital for that's where they'll get a colposcopy…when hospital A is closed right now. Oh, wait, hospital A is close? Yeah, because of COVID-19. You only take COVID-19 patients? Yes. Where are you going to send the women? To hospital A. But you're saying it's closed. Yes. They're going to be brought to the hospital. They're going to return back. Yes. And then, what? They will have to figure something else out."*

While there were challenges noted in healthcare delivery, many participants noted that community health education programs and professional training programs could be moved to an online delivery mode. Some participants even mentioned they preferred the virtual modality and planned to continue using it after the pandemic due to its reach. Several participants noted that physicians were more accepting of telemedicine appointments, which increased capacity under social-distancing protocols and will continue to do so if patients prefer virtual appointments. A tertiary-level health professional from the WPR region noted:

*"Yeah. And, especially after COVID-19 because during COVID-19, we carried out many activities only through the livestream. I mean, even for health counselling are through apps, or through live-streaming on You-Tube, etc. Now, they are more-or-less familiar with that kind of tele-scheduling."*

### Theme 6: Additional downtime due to the pandemic was advantageous in the future of programmatic implementation and CC screening potentials

Some participants mentioned that the shutdowns and downtime in implementation efforts allowed them to pivot and at times modify their efforts on targeting specific high-risk groups and planning community campaigns. A lead physician in the AFR region cited:

*"But what we then did, is we utilized that sort of downtime to plan for sort of rapid scale up post pandemic. Yeah. What we did is to actually try and mobilize resources, find out what is needed, the gloves, do you need to procure more speculums and things like that. Then to also communicate to the respective cadres within the regions that we plan to have a campaign, maybe after three months or so. We had these targets that we couldn't get. When the COVID-19 restrictions are relaxed, we are going to have a massive campaign, and we need to have massive tracking of patients."*

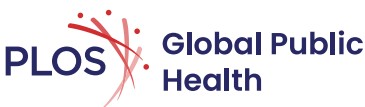

Moreover, the manner in which countries around the world united and mobilized resources to address COVID-19 demonstrated to the participants the hidden potential to address other global health disparities. Participants reported feeling more confident in improving health outcomes for CC. A tertiary-level health professional from the SEAR region elaborated:

*"Again, the COVID-19 testing only like gives us confidence. How they centralized. When the government was able to do that COVID-19 testing in such a huge manner in a very short time with limited resources. So, it should be possible for HPV DNA also, so that gives us the confidence."*

## Discussion

The emerging healthcare delivery literature has made it abundantly clear that COVID-19 had a substantial impact on health systems across the globe. On-the-ground experts included in this study perceived a shift in government priorities, which resulted in shifts in human and financial resources. Public health initiatives aimed at reducing COVID-19 transmission as well as community fear surrounding an unknown virus effectively decreased the opportunities that individuals had to go into clinics in-person to receive CC preventive services, including vaccination and screening. This effect was reported in a number of studies, who quantified vaccination, screening, and treatment rates, comparing pre-pandemic to mid-pandemic rates [6–10]. Nonetheless, the pandemic presented global healthcare systems with opportunities to find strategies to strengthen preventive services and programs. Community members gained familiarity in modes of viral transmission and ways to protect oneself, which can be used to promote CC preventive services. The laboratories were equipped with PCR machines for COVID-19 testing and an increasing number of technicians were trained to run these tests. Because these machines are not specific to COVID-19, there exists increased infrastructure for other molecular tests such as HPV testing. Molecular diagnostic companies are also optimistic in the rise of molecular testing potentials, as cited in Poljak et. al [19].

While many health systems revealed poor coordination efforts, there was a need to shift the ways in which some practices were conducted. The rise in virtual activities associated with social distancing protocols may have resulted in increased acceptance and usage of telemedicine and web-delivered trainings for the healthcare professional workforce. HPV DNA tests, in which samples can be collected without the need of a clinician, seemed to gain increased acceptance among healthcare professionals and public health decision-makers as a method to continue screening in times of health emergencies, despite the cost. Numerous studies highlighted the rise of telemedicine and self-sampling as ways that CC preventive efforts continued to function and can increase rates even post-pandemic [8,14,15,18]. The additional time experts had during programmatic holds or shutdowns allowed for increased planning efforts and strategies to target those most at risk. Ultimately, the way the global community mobilized together to tackle COVID-19 proved the ability for health systems to work together to help prevent disease transmission and outbreaks.

Some prior studies mentioned the lack of impact of COVID-19, especially on colposcopy and treatment rates [7–9]. While we did not directly find these results, we did become aware of the differential impact COVID-19 had on regions around the world. For instance, many African countries noted short-term shutdowns of health services. Moreover, studies on high-income countries expressed the inability of patients to afford healthcare services during the pandemic [14]. As our study focused primarily on LMICs, whose populations experience these hardships under non-pandemic conditions, we did not find increased inability to afford healthcare be a driving consequence of COVID-19.

This study had some limitations, which need to be acknowledged. The study sample was comprised of public health and health care practitioners who responded to email invitations among those selected by WHO regional advisors and study team members based on their expertise, knowledge, and involvement in CC screening programs in their countries. While the initial goal was to obtain representation from each WHO region, some were less responsive than others that maybe reflective of overburdened healthcare systems during the pandemic. Thus, study conclusions may not reflect experiences from individuals in countries that were not a part of the study. The data presented were validated through notes



taken during interviews and transcripts developed from audio recordings. However, we were unable to conduct participant checking to ensure our understanding of certain points was as the interviewee intended. Lastly, as the pandemic was still a global emergency impacting health systems during the time we conducted the interviews, some responses were theoretical as to how certain systems may change because of COVID-19, and future research could generate the concrete evidence to confirm these hypotheses. Further, the primary focus of our study was on screening services so when participants did comment on vaccination, precancer treatment, and cancer treatment rates, we did not probe to learn more about the impact of COVID-19 on those rates.

The study has many strengths. To the best of our knowledge, this was one of the first studies to gather global in-depth data, using a qualitative method of inquiry, in capturing the practice-based understanding for the implementation of CC screen and treatment programs. We reached a point of data saturation, which means that interviewers and analysts noted repetitions of concepts and ideas presented by the participants. This allowed us to feel confident in drawing inferences about the impact of the COVID-19 pandemic from these data. The semi-structured nature of the in-depth interviews allowed us to strengthen the interview guide as more interviews were conducted. We interviewed individuals involved in many aspects of CC screening programs, from the research up to the national policy-making level, which allowed us to gain multiple insights and perspectives on the impacts of COVID-19. Our future studies, post-pandemic, will evaluate whether these strategies have been kept in place and whether they have helped to sustain and strengthen CC screening programs across the globe.

Overall, study findings suggest that strategies that helped keep health services functional throughout the pandemic have the potential to be leveraged to improve CC prevention for rapid progress towards CC elimination.

## Supporting information

**S1 Checklist.  Inclusivity questionnaire.**
(DOCX)

## Acknowledgments

We sincerely thank the study participants for their insights and time.

## Disclaimer

Where authors are identified as personnel of the International Agency for Research on Cancer or World Health Organization, the authors alone are responsible for the views expressed in this article and they do not necessarily represent the decisions, policy or views of the International Agency for Research on Cancer or World Health Organization. The observations and conclusions in this article are those of the authors and do not represent the official position of the National Cancer Institute, the National Institutes of Health, or the US federal government. At the time of this work, authors OO and JR were employed by the University of New Mexico Comprehensive Cancer Center and author AL was a student at the University of Washington.

## Author contributions

**Conceptualization:** Patti Gravitt, Linda Eckert, Maribel Almonte, Nathalie Broutet, Prajakta Adsul.

**Data curation:** Prajakta Adsul.

**Formal analysis:** Anisha M. Loeb, Patti Gravitt, Maribel Almonte, Prajakta Adsul.

**Funding acquisition:** Prajakta Adsul.

**Investigation:** Anisha M. Loeb, Patti Gravitt, Allison Frank, Douglas M. Puricelli Perin, Kalina Duncan, Linda Eckert, Joseph Rodman, Prajakta Adsul.

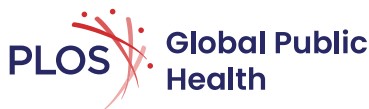

**Methodology:** Patti Gravitt, Linda Eckert, Maribel Almonte, Nathalie Broutet, Prajakta Adsul.

**Project administration:** Patti Gravitt, Linda Eckert, Maribel Almonte, Nathalie Broutet, Prajakta Adsul.

**Resources:** Maribel Almonte, Nathalie Broutet, Prajakta Adsul.

**Software:** Joseph Rodman, Prajakta Adsul.

**Supervision:** Patti Gravitt, Kalina Duncan, Maribel Almonte, Prajakta Adsul.

**Validation:** Patti Gravitt, Maribel Almonte, Joseph Rodman.

**Writing – original draft:** Anisha M. Loeb, Joseph Rodman, Prajakta Adsul.

**Writing – review & editing:** Anisha M. Loeb, Patti Gravitt, Douglas M. Puricelli Perin, Linda Eckert, Maribel Almonte, Joseph Rodman, Oyeleke A Oyebamiji, Prajakta Adsul.

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
