## [Decision Letter · Decision Letter 0]

10 May 2024

PGPH-D-24-00517

Assessing the Global Implications of the COVID-19 Pandemic on Cervical Cancer Screening Initiatives: A Comprehensive Evaluation

Dear Dr. Adsul,

Thank you for submitting your manuscript to PLOS Global Public Health. After careful consideration, we feel that it has merit but does not fully meet PLOS Global Public Health’s publication criteria as it currently stands. Therefore, we invite you to submit a revised version of the manuscript that addresses the points raised during the review process.

We look forward to receiving your revised manuscript.

Kind regards,

Sadeep Shrestha

Academic Editor

Journal Requirements:

2. Please provide additional details regarding participant consent. In the ethics statement in the Methods and online submission information, please ensure that you have specified (1) whether consent was informed and (2) what type you obtained (for instance, written or verbal, and if verbal, how it was documented and witnessed).”

3. Please include a complete copy of PLOS’ questionnaire on inclusivity in global research in your revised manuscript. Our policy for research in this area aims to improve transparency in the reporting of research performed outside of researchers’ own country or community. The policy applies to researchers who have travelled to a different country to conduct research, research with Indigenous populations or their lands, and research on cultural artefacts. The questionnaire can also be requested at the journal’s discretion for any other submissions, even if these conditions are not met. Please find more information on the policy and a link to download a blank copy of the questionnaire here: https://journals.plos.org/globalpublichealth/s/best-practices-in-research-reporting. Please upload a completed version of your questionnaire as Supporting Information when you resubmit your manuscript.

Additional Editor Comments (if provided):

Reviewers' comments:

Reviewer's Responses to Questions

**Comments to the Author**

1. Does this manuscript meet PLOS Global Public Health’s publication criteria ? Is the manuscript technically sound, and do the data support the conclusions? The manuscript must describe methodologically and ethically rigorous research with conclusions that are appropriately drawn based on the data presented.

Reviewer #1: Yes

Reviewer #2: Partly

2. Has the statistical analysis been performed appropriately and rigorously?

Reviewer #1: N/A

Reviewer #2: N/A

3. Have the authors made all data underlying the findings in their manuscript fully available (please refer to the Data Availability Statement at the start of the manuscript PDF file)?

Reviewer #1: No

Reviewer #2: No

4. Is the manuscript presented in an intelligible fashion and written in standard English?

Reviewer #1: Yes

Reviewer #2: Yes

5. Review Comments to the Author

Reviewer #1: Thank you for the opportunity to review! This paper presents important findings on how healthcare systems worldwide responded to the COVID-19 pandemic, illustrating how healthcare systems in LMICs faced different challenges from those in HIC. These results will be a valuable contribution to the literature on emergency preparedness, specifically relating to challenges in managing chronic diseases in low-resource settings.

Reviewer #2: Interesting and timely manuscript resulting in first-hand accounts from around the globe of how the pandemic impacted cervical cancer prevention services. It is nice to see that the authors present opportunities as well as challenges that may help guide future research. Detailed comments below.

Abstract:

“supporting countries’ abilities to mobilize resources.”

-In the methods, it would be useful to have some details about the simultaneous translation (in person? AI?) and/or number of languages the interviews took place in (X in English, X in other languages?)

-In methods, although the participant sample is described, additional details that speak to sample representatives should be included. It’s a qualitative study, but it’s still helpful to know how many participants of how many categories/per country were included.

-Also methods: The sample of countries requires more description at least in terms of income (for example, X low income countries, X low-middle, X upper middle, etc.). Would be useful to know how many countries were included and which ones/how many of those have existing HPV screening and/or vaccination programs (as opposed to pilot or research projects) and since when (or at least giving a range of years or older vs. new?). This is important contextual information that should be discussed within each analysis section as we can expect the impact of the pandemic to be different on established programs vs. research/pilot projects and income level of each country. For example, some themes may be more relevant to specific countries/regions with more resources (like telemedicine) and that should be acknowledged.

-In general, conclusions are over-drawn. Given the methods and sample size and selection (overall and per country) results can be categorized as exploratory at best. For example, in Lines 200-216, one person’s opinion does not mean that is actually the case. Interpretation should be revised from “improvement in awareness among the populations was an important advantage” to “may be a potential advantage”. This is the case throughout the manuscript.

-In Results, it would be important to know which themes were most/least common and which were most/least often associated with particular regions/countries (maybe a table?). It would also be good to know if different themes were most often mentioned or were considered more important by different types of interviewees (ministry officials, providers, program managers, etc.).

Line 143-146: discusses VIA as the screening method, not HPV testing, which is confusing given the abstract which specifically refers to HPV testing. Clarifying screening methods used in the different countries would be helpful, as these two methods are expected to present at least some different challenges as a result of COVID.

Line 148-152: same issue as above, would be helpful to clarify the context in which this was noted (a country with an existing HPV program but no self-sampling? Introduction of HPV as a result of the pandemic?)

Line 167-168: “the poorly-funded programs”. This also assumes that all countries involved were in low-resource-settings, but that is not clear from the methods which discusses a range of global regions. In any case, LMICs will vary greatly in available resources and some may have much funded programs than others.

Line 234-235: incidence already slowed down

Line 259-260: Not clear why fragmented healthcare systems are crucial for screening?

Line 320-321: it’s curious that COVID misinformation and vaccine hesitancy were not mentioned in the interviews, since that has been an often-reported outcome of the pandemic in various countries including LMICs. Perhaps those folks have less contact with the health system and that skews the perception. But the literature on these issues should be acknowledged.

Lines 328-339: Again, some caution in interpreting results from these interviews is needed here, firm conclusions cannot be drawn from the data.

-Remove commas in Lines 45, 50, 93, 175, 275, 301, 338

6. PLOS authors have the option to publish the peer review history of their article (what does this mean? ). If published, this will include your full peer review and any attached files.

**Do you want your identity to be public for this peer review?** For information about this choice, including consent withdrawal, please see our Privacy Policy .

Reviewer #1: No

Reviewer #2: No

---

## [Decision Letter · Decision Letter 1]

24 Jan 2025

PGPH-D-24-00517R1

Assessing the Global Implications of the COVID-19 Pandemic on Cervical Cancer Screening Initiatives: A Comprehensive Evaluation

Dear Dr. Adsul,

Thank you once again for submitting your revised manuscript to PLOS Global Public Health and for your patience during the unprecedented delays. The reviewers have reviewed your revised draft and have suggested further recommendations. Therefore, we invite you to carefully review their recommendations and submit a revised version of the manuscript that addresses the points raised during the review process.

We look forward to receiving your revised manuscript.

Kind regards,

Edina Amponsah-Dacosta, Ph.D., MPH

Academic Editor

Journal Requirements:

Reviewers' comments:

Reviewer's Responses to Questions

**Comments to the Author**

1. If the authors have adequately addressed your comments raised in a previous round of review and you feel that this manuscript is now acceptable for publication, you may indicate that here to bypass the “Comments to the Author” section, enter your conflict of interest statement in the “Confidential to Editor” section, and submit your "Accept" recommendation.

Reviewer #2: All comments have been addressed

Reviewer #3: All comments have been addressed

2. Does this manuscript meet PLOS Global Public Health’s publication criteria ? Is the manuscript technically sound, and do the data support the conclusions? The manuscript must describe methodologically and ethically rigorous research with conclusions that are appropriately drawn based on the data presented.

Reviewer #2: Yes

Reviewer #3: Yes

3. Has the statistical analysis been performed appropriately and rigorously?

Reviewer #2: N/A

Reviewer #3: Yes

4. Have the authors made all data underlying the findings in their manuscript fully available (please refer to the Data Availability Statement at the start of the manuscript PDF file)?

Reviewer #2: Yes

Reviewer #3: Yes

5. Is the manuscript presented in an intelligible fashion and written in standard English?

Reviewer #2: Yes

Reviewer #3: Yes

6. Review Comments to the Author

Reviewer #2: This is a timely and important manuscript that highlights challenges, but more significantly, opportunities to expand and scale-up cervical cancer prevention efforts by leveraging changes stemming from the COVID pandemic. The sample is diverse, the analysis is rigorous, and the conclusions are well-connected to the data.

There are a few areas where clearer writing or adding an explanatory sentence would be beneficial. I also appreciate the desire to connect the manuscript to the implementation literature, but discussing the framework used in the parent project is confusing (I was expecting to see more of it throughout the results/conclusions) and not immediately relevant to these findings. Specific comments are on the margins of the PDF.

Reviewer #3: Comments and Recommendations:

1. The introduction provides a clear context about the pandemic's impact on cervical cancer screening but could better emphasize the novelty of this study compared to existing research. Highlight specific gaps in the current literature that this study aims to address.

Recommendations: Add a paragraph summarizing how the study uniquely contributes to understanding the global implications of COVID-19 on cervical cancer screening.

2. While six themes are identified, some sections, such as Theme 2 (Fear of contracting COVID-19) and Theme 5 (Fragmentation of healthcare systems), need more robust linking to the findings' broader implications.

Recommendations: Expand on how these themes can guide policymakers in strengthening healthcare systems for future resilience.

3. The manuscript describes using a grounded theory approach but does not detail how inter-coder reliability was ensured or how saturation was determined.

Recommendations: Include a brief explanation of inter-coder reliability measures and the method used to determine thematic saturation.

4. The limitations section is thorough but lacks emphasis on potential biases due to the reliance on WHO-selected participants.

Recommendations: Explicitly mention how participant selection might skew results and suggest strategies to mitigate this in future research.

5. The discussion section could better integrate the findings with actionable recommendations for programmatic improvements post-pandemic.

Recommendations: Propose specific strategies for leveraging pandemic-induced infrastructure improvements (e.g., molecular testing platforms for HPV screening) in LMICs.

7. PLOS authors have the option to publish the peer review history of their article (what does this mean? ). If published, this will include your full peer review and any attached files.

**Do you want your identity to be public for this peer review?** For information about this choice, including consent withdrawal, please see our Privacy Policy .

Reviewer #2: No

Reviewer #3: **Yes: ** ALLTALENTS T MURAHWA

---

## [Editor Report · Decision Letter 2]

28 Feb 2025

Assessing the Global Implications of the COVID-19 Pandemic on Cervical Cancer Screening Initiatives

PGPH-D-24-00517R2

Dear Dr. Adsul,

We are pleased to inform you that your manuscript 'Assessing the Global Implications of the COVID-19 Pandemic on Cervical Cancer Screening Initiatives' has been provisionally accepted for publication in PLOS Global Public Health.

Best regards,

Edina Amponsah-Dacosta, Ph.D., MPH

Academic Editor